# Mode-multiplexing deep-strong light-matter coupling

Joshua Mornhinweg [1,2], Laura Katharina Diebel[1], Maike Halbhuber[1], Michael Prager[1], Josef Riepl[1], Tobias Inzenhofer [1], Dominique Bougeard[1], Rupert Huber [1] ✉ & Christoph Lange [2] ✉

Dressing electronic quantum states with virtual photons creates exotic effects ranging from vacuum-field modified transport to polaritonic chemistry, and squeezing or entanglement of modes. The established paradigm of cavity quantum electrodynamics maximizes the light-matter coupling strength $\Omega_R/\omega_c$, defined as the ratio of the vacuum Rabi frequency and the frequency of light, by resonant interactions. Yet, the finite oscillator strength of a single electronic excitation sets a natural limit to $\Omega_R/\omega_c$. Here, we enter a regime of record-strong light-matter interaction which exploits the cooperative dipole moments of multiple, highly non-resonant magnetoplasmon modes tailored by our metasurface. This creates an ultrabroadband spectrum of 20 polaritons spanning 6 optical octaves, calculated vacuum ground state populations exceeding 1 virtual excitation quantum, and coupling strengths equivalent to $\Omega_R/\omega_c = 3.19$. The extreme interaction drives strongly subcycle energy exchange between multiple bosonic vacuum modes akin to high-order non-linearities, and entangles previously orthogonal electronic excitations solely via vacuum fluctuations.

Strong coupling of light and matter is governed by the exchange of energy via virtual excitations, at the Rabi frequency, $\Omega_R$. In the ultrastrong[1–16] ($0.1 \leq \Omega_R/\omega_c < 1$) and deep-strong[7,9,14] coupling (DSC) regime ($\Omega_R/\omega_c \geq 1$), the vacuum ground state is characterised by an unusually large population of virtual excitations[17,18] with non-classical properties, including significant single or two-mode squeezing[17,18]. This results in spectacular effects of cavity quantum electrodynamics (c-QED)[19] including the vacuum Bloch-Siegert shift[10], polaritonic chemistry[20–22], the creation of photon-bound excitons[23], vacuum-field induced charge transport[13,15], strong nonlinearities[24], tunnelling[25] and coherent polariton scattering[26]. Landau polaritons[4,5,7,27] of plasmonic THz resonators were the first optical systems to enter the DSC regime[7,9,14] with $\Omega_R/\omega_c = 1.43$ and a large ground state population of 0.37 virtual photons[7]. Moreover, they enable non-adiabatic switching[28] of the extremely squeezed quantum vacuum ground state, which may enable the observation of Unruh-Hawking radiation[29,30] with a further boost of $\Omega_R/\omega_c$. While previous investigations have demonstrated remarkable progress in this direction, they also discovered fundamental barriers given by light-matter decoupling[31] or dissipation[32]. One of the most significant restrictions for increasing $\Omega_R/\omega_c$, however, is the paradigm of resonant light-matter interaction of a single photonic and a single electronic mode, limiting the maximum contributing oscillator strength (Fig. 1a).

Here, we present a conceptually different approach to DSC which overcomes the limitations of resonant coupling and establishes record coupling strengths: for sufficiently large spatial overlap of light and matter polarisation fields, even electronic excitations that are strongly detuned from the optical mode can substantially boost the vacuum ground state population. We leverage this idea by multiplexing the interaction, exploiting multiple, non-resonant plasmon modes which simultaneously and cooperatively couple with extreme strength to several optical modes of a metallic metasurface (Fig. 1b). Our resonator custom-tailors the plasmon quantum states, while significantly reducing the cavity size as compared to previous approaches, and provides

[1]Department of Physics, University of Regensburg, 93040 Regensburg, Germany. [2]Department of Physics, TU Dortmund University, 44227 Dortmund, Germany. ✉e-mail: rupert.huber@ur.de; christoph.lange@tu-dortmund.de

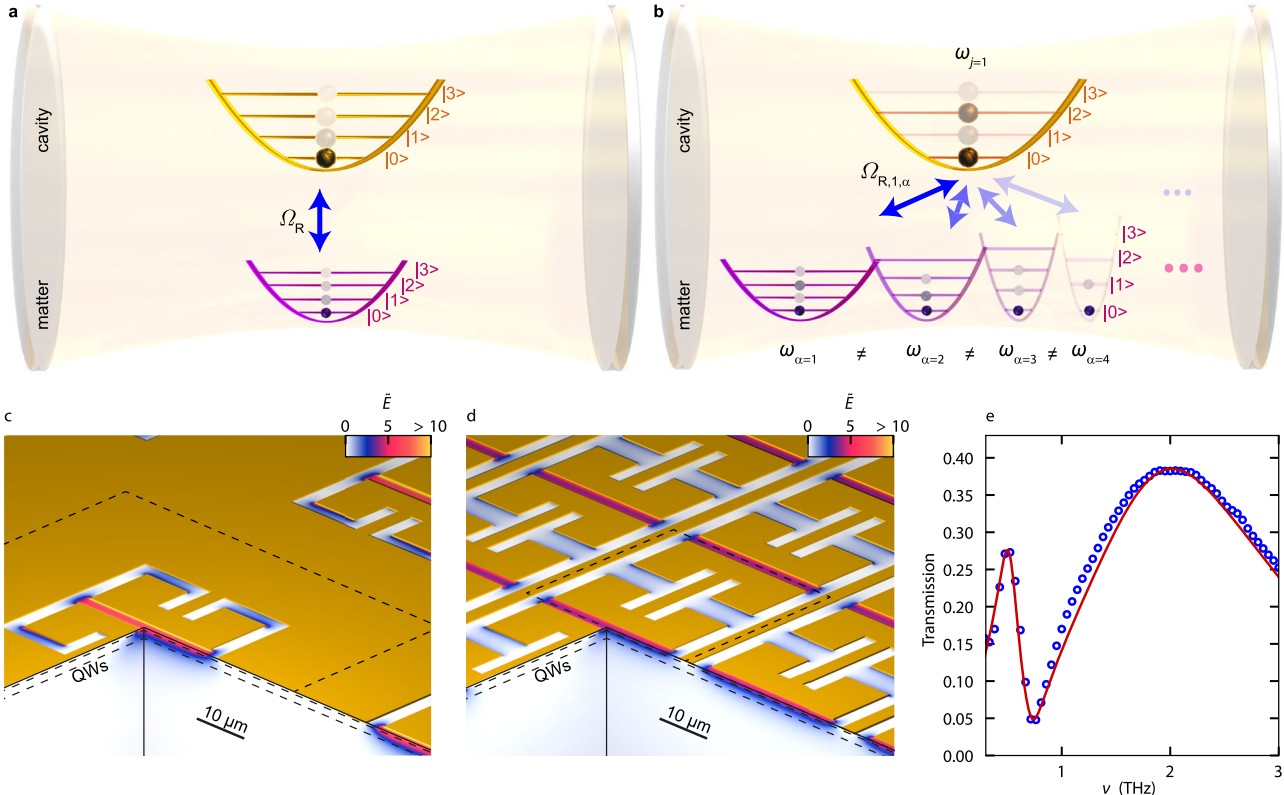

**Fig. 1 | Multi-mode light-matter coupling and ultracompact metasurface.**
**a** Illustration of resonant ultrastrong coupling of a single cavity mode (upper parabola) to a single matter excitation (bottom parabola) with a vacuum Rabi frequency $\Omega_R$. The weak population by virtual excitations in the vacuum ground state is indicated by semi-transparent spheres. **b** Illustration of deep-strong coupling of one light mode (upper parabola) with a frequency of $\omega_{j=1}$ to multiple matter excitations (bottom parabolas) with frequencies $\omega_\alpha$, by vacuum Rabi frequencies $\Omega_{R,j,\alpha}$ under off-resonant conditions. Owing to the extremely large light-matter coupling, a significant number of virtual excitations are present. **c** Three-dimensional cut-away illustration of a conventional metasurface structure (gold shape) and its electric near-field distribution $\tilde{E}$, for the fundamental LC mode at $\nu_1 = 0.52$ THz. The unit cell (size: $60\,\mu m \times 60\,\mu m$) is indicated by the dashed line. QW: quantum well stack. **d** Highly compacted metasurface with a unit cell size of $30\,\mu m \times 32.5\,\mu m$. **e** Measured THz transmission of the bare resonator array (blue circles), and calculation (red curve).

large near-field enhancement. The resulting strong subcycle transfer of vacuum energy between optical and electronic modes leads to an ultrabroadband spectrum of more than 20 distinct, extremely strongly coupled resonances distributed over up to 6 optical octaves. The vacuum ground state is populated by up to 1.17 virtual photons, which leads to significant squeezing and corresponds to a record coupling strength of $\Omega_R^s/\omega_s = 3.19$ for a hypothetical single pair of light and matter modes, for reference. Surpassing previous values almost two-fold, this opens up unique possibilities for a range of non-adiabatic c-QED phenomena such as vacuum-field modified transport[13,15], control of cavity chemistry[20–22], and Unruh-Hawking radiation[29,30]. Moreover, our cavity mediates extremely strong coupling of plasmons that are orthogonal in their bare state - a distinct hallmark of the regime of deep-strong multi-mode coupling, which may in the future be exploited to drive phase transitions[33] in quantum materials, solely by the vacuum field.

## Results

Our resonator design capitalises on the principles of subwavelength near-field confinement[34,35] of THz vacuum fields and current distributions in metasurfaces[36], exploiting their great flexibility for designing the spectral shape, resonance frequencies, and near-field structure of optical modes. The structures are specifically optimised for multi-mode coupling and address the two central challenges of our c-QED scenario: designing a broadband spectrum of multiple distinct electronic modes to boost the coupling strength as well as reducing the unit cell size multifold over existing designs. The metasurface is inspired by an inverted layout[37] (Fig. 1c). As a key advance, we discard the thought pattern of individual, spatially isolated resonators which are generously separated to avoid nearest-neighbour couplings. Instead, we perform finite-element frequency-domain (FEFD) calculations of the current distribution within the metal layer to identify and remove areas of low current density, which are irrelevant to the performance of the resonator. In addition, we exploit the symmetries of the structure to merge adjacent current paths of opposite phase, allowing us to eliminate the affected elements entirely (see Supplementary Information). The resulting metasurface (Fig. 1d) is highly compact and, as a result of the changed topology, can no longer be separated into individual resonators nor strictly be classified as an inverse or a direct design. More precisely, its unit cell is approximately only a quarter as large as compared to conventional designs[4,7], while a similar near-field enhancement is achieved. Its $x$-polarised modes include the fundamental LC resonance with a centre frequency of $\nu_{j=1} = 0.52$ THz, while a higher-order mode lies at $\nu_{j=2} = 1.95$ THz (Fig. 1e), with $j$ denoting the cavity mode index.

We experimentally evaluate our multi-mode concept by fabricating the gold resonators by electron-beam lithography on top of n-doped GaAs multi-quantum well (QW) heterostructures which host two-dimensional electron gases with a nominal charge carrier concentration of $\rho_{QW} = 1.8 \times 10^{12}\,cm^{-2}$, and are separated by AlGaAs barrier layers[7] (see Methods). A magnetic bias field **B** of up to 5.5 T is oriented perpendicularly to the QW plane and Landau-quantises the electrons, achieving a cyclotron resonance (CR) with a tuneable frequency $\nu_c = e|\mathbf{B}|/2\pi m^*$, where $e$ is the elementary charge, $m^* = 0.07\,m_e$ is the

electron effective mass and $m_e$ is the free electron mass. Owing to the comparably large effective mass of electrons in gold of $m_{Au}^*/m_e \gtrsim 1$ (ref. 38), the CR of electrons in the resonator material itself is neglected.

In the assembled structure, the periodicity of the field localisation of the metasurface quantises the wave vectors of the light field that can couple to the electrons in the QWs (Fig. 2a). As a result, the mode's vacuum field couples to a fan of distinct plasmon resonances[32]. The plasmons feature a frequency of $\omega_P(\mathbf{q}_x) = \sqrt{\frac{\rho_{2D}e^2}{2m^*\epsilon_0\epsilon_{eff}(\mathbf{q})}|\mathbf{q}_x|}$ (refs. 39,40) (Fig. 2a, red graph), where $\rho_{2D}$ is the total carrier density integrated

over all QWs, and $\epsilon_0$ and $\epsilon_{eff}(\mathbf{q})$ are the vacuum permittivity and effective dielectric function, respectively. The plasmon wave vector $\mathbf{q}_x(\alpha) = \frac{2\pi}{L_x}\alpha$ depends on the unit cell size in $x$-direction, $L_x$ and the mode index, $\alpha \in \mathbb{Z}$ (see Supplementary Information for the full theory). The small value of $L_x = 30\,\mu m$ favourably increases the energy spacing as compared to previously investigated structures with larger unit cells[7], leading to more distinct resonances. Plasmon excitation is accounted for up to a maximum index $|\alpha_c| = 10$, beyond which the near-field amplitude is negligible. Linear combinations of plasmon pairs $(-\mathbf{q}_x, \mathbf{q}_x)$ form bright and dark standing-wave modes which hybridise

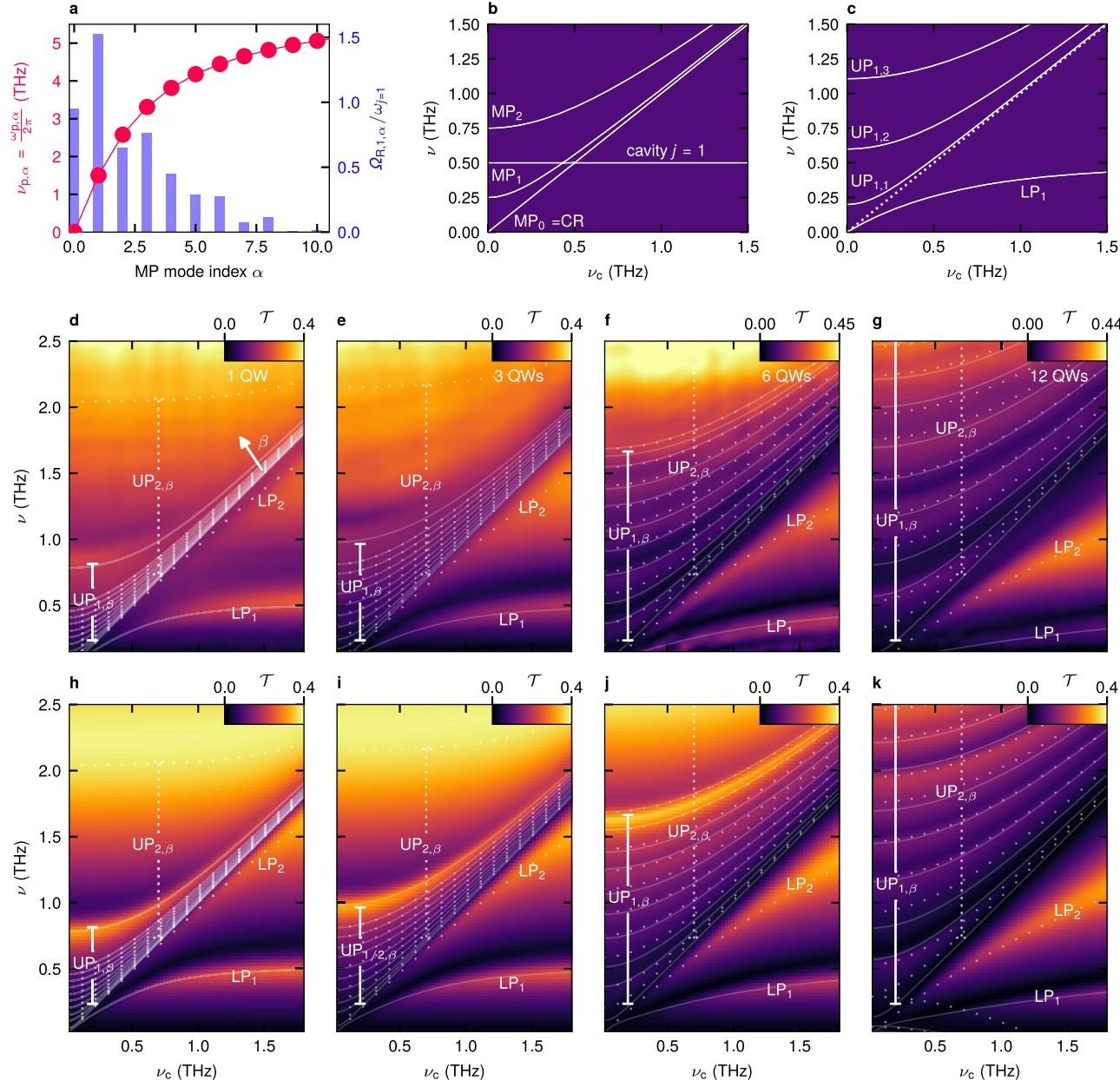

**Fig. 2 | Deep-strong light-matter coupling. a** Plasmon frequency $\nu_{p,\alpha}$ and coupling strength $\Omega_{R,1,\alpha}/\omega_{j=1}$ for each MP (magnetoplasmon) mode $\alpha$ in the case of the sample with 48 QWs (quantum wells). **b**, **c** Illustration of multi-mode coupling of one cavity mode to several matter modes, $MP_0 = CR$, $MP_1$ and $MP_2$, as a function of the cyclotron frequency, $\nu_c$. **b** uncoupled modes. **c** Coupled modes comprising of one lower polariton ($LP_1$) and three upper polaritons ($UP_{1,1}$, $UP_{1,2}$ and $UP_{1,3}$). **d** THz magneto-transmission $\mathcal{T}$ as a function of $\nu_c$ of the single-QW structure. The continuous white curves trace the polariton modes obtained from the multi-mode

Hopfield model for the first resonator mode (coupling strength: $\eta_1 = 0.55$). The dotted white curves represent the polaritons linked to the higher mode $\nu_2$ ($\eta_2 = 0.13$). **h** Spectrum obtained from time-domain quantum model and identical polariton frequencies, for comparison. **e** Transmission of the 3-QW structure ($\eta_1 = 0.76$) and **i** simulation. **f** Transmission of the 6-QW structure ($\eta_1 = 1.34$), and **j** simulation. **g** Transmission of the 12-QW structure ($\eta_1 = 2.32$), and **k** simulation.

with the CR at $\omega_c = 2\pi\nu_c$, forming $2|\alpha_c| + 1 = 21$ magnetoplasmon (MP) resonances (see Supplementary Information). Each MP with a frequency of $\omega_{MP,\alpha} \propto \sqrt{\omega_{P,\alpha}^2 + \omega_c^2}$ (ref. [41]; Fig. 2b) couples to the resonator modes, $j$, with individual vacuum Rabi frequencies $\Omega_{R,j,\alpha}$ depending on the MP amplitude (Fig. 2a, blue bars) and the detuning, $\nu_j - \omega_{MP,\alpha}/2\pi$. For any given value of $\nu_c$, the detuning vanishes only for a single MP mode at a time (Fig. 2b).

The coupling of the multiple plasmon modes to the electromagnetic modes of our resonator results in a set of multiple light-matter coupled modes. Each of these modes is a characteristic superposition of all MP modes and, since the cavity modes are almost orthogonal[16], only a single cavity mode. We thus reuse the cavity mode index $j$ and introduce the index $\beta$ for the resulting 22 magnetoplasmon cavity polaritons (MPPs). These are categorised into a lower polariton, $LP_{j,(\beta=0)}$, and, as a signature of multi-mode coupling, several upper polaritons, $UP_{j,1 \leq \beta \leq \alpha_c+1}$. The presence of only one LP mode, but several UP modes is the fingerprint of cooperative coupling of all independent MP modes to one common cavity mode. Dark modes ($\beta < 0$) are included in our model but not further discussed here (see Supplementary Information). As shown in the simplified example in Fig. 2c, all MPP frequencies rise monotonically when $\nu_c$ is increased from zero. The $LP_{j=1}$ asymptotically emerges from below the CR (dotted line) near $\nu_c \approx 0$ and converges towards the cavity frequency for large $\nu_c$. The $UP_{j=1,\beta}$ modes, for $\nu_c = 0$, start with a finite frequency, which is defined by the plasma frequencies of the constituent MPs as well as the diamagnetic shift caused by the photon field[17]. All UP modes bend upwards with increasing $\nu_c$ and asymptotically converge towards the CR.

In a first experimental campaign, we investigate this distinct regime of multi-mode coupling for four samples containing 1, 3, 6 and 12 QWs, respectively, by THz time-domain magneto-spectroscopy at cryogenic temperatures as a function of $\nu_c$ (see Methods). For the single-QW structure (Fig. 2d), the single $LP_1$ mode is located below the CR for vanishing $\nu_c$ and approaches the frequency of the bare first cavity mode at 0.5 THz, as $\nu_c$ is increased. For $\nu_c = 0$, the $UP_{1,\beta}$ are observed as a dense fan of partially overlapping modes distributed from $\sim 0$ THz upwards. The most strongly visible UP mode, highlighted by the uppermost semi-transparent white curve, is highest in frequency with $\nu = 0.78$ THz. Increasing $\nu_c$, the entire UP mode structure curves upwards in frequency as discussed for Fig. 2c and occupies the increasingly smaller spectral bandwidth between the CR and the highest-energy UP mode. Similarly, the $LP_2$ mode related to the second cavity mode (lowermost dotted curve) branches off below the CR as $\nu_c$ is increased and reaches $\nu = 1.65$ THz at $\nu_c = 1.9$ THz. The $UP_{2,\beta}$ ensemble (upper dotted curves) is dominated by a single spectral feature centred near $\nu = 2.0$ THz for $\nu_c = 0$ THz while slightly shifting upwards with increasing $\nu_c$.

For $n_{QW} = 3$, the overall electronic dipole moment is boosted and the frequency spacing $\omega_P(\alpha)$ of the uncoupled MPs and, as a result, that of the coupled MPPs is increased (Fig. 2e). For $\nu_c = 0$ THz, the $UP_{1,\beta}$ extend from $\sim 0$ to 1.0 THz and form clearly separated resonances, evidencing the multi-mode character of the interaction. The increased coupling likewise shifts the frequencies of the $UP_{2,\beta}$ modes further up while lowering the frequencies of the $LP_j$ modes. These trends continue for $n_{QW} = 6$ (Fig. 2f), where for $\nu_c = 0$ THz, the MPP fan manifests in well-separated $UP_{1,\beta}$ modes which extend up to 1.65 THz, whereas the $UP_{2,\beta}$ reach up to 2.5 THz. Yet, owing to the coupling of all MP to the same common cavity mode, only one LP forms for each cavity mode. Implementing $n_{QW} = 12$ (Fig. 2g), we observe distinct $UP_{1,\beta}$ and $UP_{2,\beta}$ branches at frequencies reaching 2.5 THz and 3.2 THz, several of which are simultaneously ultrastrongly coupled with centre frequencies reaching up to $\approx 5$ times the bare cavity frequency, $\nu_{j=1} = 0.52$ THz.

Such extremely strong, off-resonant coupling of multiple light and matter modes with highly disparate frequencies represents an advanced setting of c-QED in which strongly subcycle exchange of vacuum energy between the coupled modes is expected. For a quantitative understanding, we developed a multi-mode theory of DSC based on the following Hopfield Hamiltonian[27,42]:

$$\hat{\mathcal{H}} = \sum_j \hbar\omega_j \hat{a}_j^\dagger \hat{a}_j + \sum_\alpha \hbar\omega_{MP}(\alpha) \hat{b}_\alpha^\dagger \hat{b}_\alpha + \sum_{\alpha,j} \hbar\Omega_{R,j,\alpha} \left( \hat{a}_j^\dagger + \hat{a}_j \right) \left( \hat{b}_\alpha^\dagger + \hat{b}_\alpha \right)$$
$$+ \sum_{\alpha,j} \frac{\hbar\Omega_{R,j,\alpha}^2}{\omega_{MP}(\alpha)} \left( \hat{a}_j^\dagger + \hat{a}_j \right)^2 + \hat{\mathcal{H}}_{ext}.$$

$$(1)$$

The first two terms describe the bare cavity and MP resonances with frequencies $\omega_j = 2\pi\nu_j$ and $\omega_{MP}(\alpha)$ and bosonic annihilation operators $\hat{a}_j$ and $\hat{b}_\alpha$, respectively. The third term describes their mutual coupling, quantified by a set of vacuum Rabi frequencies, $\Omega_{R,j,\alpha}$, and the fourth term represents the diamagnetic cavity blue shift while $\hat{\mathcal{H}}_{ext}$ implements coupling to external fields.

The individual vacuum Rabi frequencies, $\Omega_{R,j,\alpha}$, are chosen proportional to the near-field amplitude component at the corresponding MP wave vector obtained by Fourier transform of the cavity field, and are multiplied by a common factor $\lambda = E_j/\sqrt{V_j}$ which scales all $\Omega_{R,j,\alpha}$ with the amplitude $E_j$ and the volume $V_j$ of each cavity mode[5] and is adjusted for best agreement with the experiment. We include the decay of the near-field amplitude in growth direction, thereby accounting for the coupling to each QW individually (Supplementary Information). Solving Heisenberg's equations of motion for our Hamiltonian (see Methods) unveils the temporal evolution and, by Fourier transform, the spectral shape of the coupled modes. The theory accurately reproduces the measured spectra (Fig. 2h–k) regarding the line widths and oscillator strengths of all modes over the complete experimentally accessible spectral range from <0.1 THz to 6 THz, with quantitative precision. With multiple matter modes off-resonantly coupled to multiple light modes with extreme strengths, the notion of the anti-crossing point that maximises the coupling strength of a single pair of modes becomes irrelevant. More specifically, when $\Omega_{R,j,\alpha}/\omega_{MP}(\alpha) \approx 1$ is simultaneously reached for several MP modes $\alpha$, the field fluctuations of off-resonant modes influence the vacuum ground state $|G\rangle$ only slightly less than resonant ones (see Supplementary Information). In this setting, the number of virtual photons $\langle N_j \rangle = \langle G|\hat{a}_j^\dagger \hat{a}_j|G\rangle$ of each photonic mode $j$ is the appropriate figure of merit of DSC. Moreover, the relaxed resonance criterion and the coupling to common cavity modes allows for the total vacuum photon population $\sum_j \langle N_j \rangle$ to be increased almost arbitrarily by adding electronic oscillator strength within a spectral window up to several octaves wide.

For our 1-QW structure, we calculate $\langle N_1 \rangle = 0.07$ and $\langle N_2 \rangle = 4 \times 10^{-3}$. For comparison with previous demonstrations, the hypothetical coupling strength of a single pair of resonant light and matter modes that yields an equivalent vacuum photon number, $\eta_j = \Omega_{R,j}^s/\omega_j$, amounts to $\eta_1 = 0.55$ and $\eta_2 = 0.13$ for the first two cavity modes. The calculation for the 3-QW structure results in an increased number of virtual photons and effective coupling strength for the cavity modes (Tab. 1). The model also reproduces the transition from the merged ensemble of $UP_{1,\beta}$ modes (Fig. 2h) to clearly distinguishable individual MPP resonances (Fig. 2i–k). In addition, the calculation replicates the data for $n_{QW} = 6$ (Tab. 1). For $n_{QW} = 12$ we reach $\langle N_1 \rangle = 0.76$, $\langle N_2 \rangle = 0.08$ and $\eta_1 = 2.32, \eta_2 = 0.60$. While these values already exceed previous records significantly[7,14], we push the limits of our approach even further with two additional structures employing 24 and 48 QWs, respectively. With the comparably large extension of

these QW stacks in the $z$ direction of up to ~2 μm, the lowermost QWs are located at a distance from the surface where the near-field amplitude of the bare cavity is significantly reduced relative to the plane of the resonators. However, owing to superradiant coupling prevalent at high densities of Landau electrons[43], the polarisation components of all Landau electrons throughout the QW stack are synchronised in phase to a high degree. The optical mode thus couples to this collective polarisation field of all QWs in the entire stack, increasing the effective mode volume. While the coupling strength is hereby reduced, its increase by the boosted net oscillator strength by far dominates. Both effects are included in our FEFD model. The deep-strongly coupled polariton modes of these optimised structures extend over an even wider frequency range and create a highly structured spectrum (Fig. 3a,c). The increased interaction strength moreover causes a small signature of coupling of $x$ and $y$-polarised components by the gyrotropic nature of the CR. As a result, we observe an additional, faint polariton branch located ~200 GHz above the LP$_2$ mode for $\nu_c = 2$ THz, which can be attributed to a $y$-polarised resonator mode and is invisible for our $x$-polarised probe pulses at lower coupling strengths.

Our calculations (Fig. 3b,d) confirm yet higher vacuum photon populations and coupling strengths for $n_{QW} = 24$ (Tab. 1), finally achieving $\langle N_1 \rangle = 1.00$, $\langle N_2 \rangle = 0.17$ as well as $\eta_1 = 2.83, \eta_2 = 0.88$ for $n_{QW} = 48$. Here, for the first time, the vacuum photon population of a single coupled optical mode reaches unity, while the combined ground state population of both modes, $\langle N \rangle = \langle N_1 \rangle + \langle N_2 \rangle = 1.17$, comfortably surpasses it. The latter value would be found for an effective coupling strength of $\Omega_R^s/\omega_s = 3.19$ – exceeding existing structures with $\eta = 1.43$ (ref. 7) and $\eta = 1.83$ (ref. 14) by almost a factor of 2. The strong back-action of the cavity vacuum fields moreover leads to a combined population of the MPs by 1.06 virtual excitations, and results in strong mixing of the formerly orthogonal matter modes. This

perspective holds out the prospect of wide-ranging control of transport[13,15], chemical reactions[20–22] or phase transitions[33], merely by vacuum fluctuations.

The unconventional nature of this extremely strong multi-mode mixing is especially evident when the subcycle dynamics are investigated directly in the time-domain. To this end, we analyse the polarisation dynamics of our structure by the transmitted THz field of the 48-QW sample. At $\nu_c = 0.52$ THz, the experimental waveform consists of a pronounced initial cycle at a delay time of $t = 0$ ps (Fig. 4a). From $t = 0.5$ ps onwards, rapid trailing oscillations are observed which exhibit beating patterns (indicated by arrows) resulting from the rapid energy exchange between multiple light and matter modes. The spectrum has a global maximum located near 2 THz and several small, adjacent local maxima resulting from individual MPP modes (Fig. 4a, inset). Calculating the internal dynamics by our time-domain theory we find that the electric field of the first cavity mode (Fig. 4b, black curve) oscillates on time scales much faster than the cycle duration of the bare mode, $(\nu_1)^{-1}$, (Fig. 4b, grey-shaded area). Its spectrum features 8 distinct local maxima of comparable amplitude located between 0.2 THz, where the LP$_1$ is situated, and up to 4.5 THz (Fig. 4b, inset). Exceeding the frequency $\nu_1$ of the uncoupled cavity by almost an order of magnitude, these features directly result from subcycle energy transfer driven by extreme light-matter coupling strengths, $\Omega_{R,j=1,\alpha}$. Further non-vanishing contributions extend to as low as 0.05 THz and up to 6 THz for a total spectral bandwidth of more than 6 optical octaves. Similarly, the polarisation of the first MP mode ($\nu_{MP,\alpha=1} = 1.2$ THz) is strongly structured by multiple oscillations (Fig. 4c). Its spectrum is characterised by four main contributions located between 0.18 and 2.59 THz, as well as further, less pronounced components at higher frequencies (Fig. 4c, inset). Moreover, since the large coupling strengths $\Omega_{R,j=1,\alpha}$ to the same cavity mode cause all MP modes to

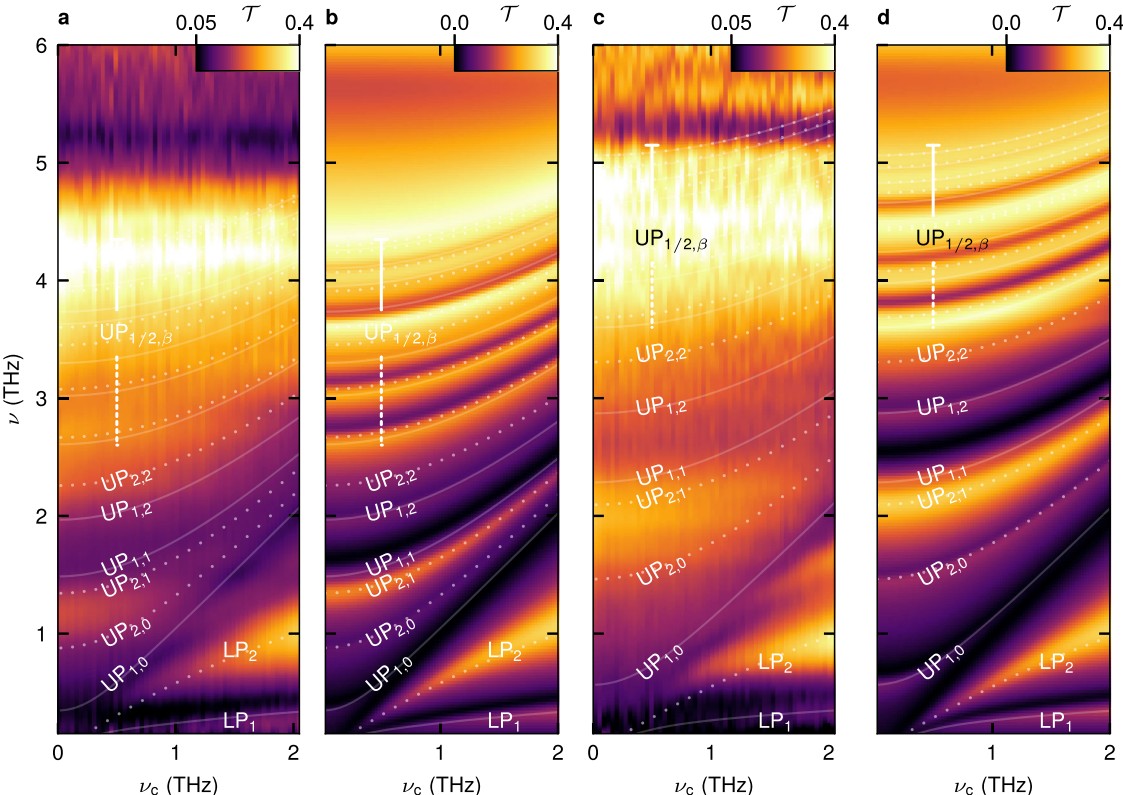

**Fig. 3 | Extremely strong, multi-octave light-matter coupling. a** THz magneto-transmission $\mathcal{T}$ of the 24-QW sample as a function of the cyclotron frequency, $\nu_c$ (see Fig. 2). The extended Hopfield model yields coupling strengths of $\eta_1 = 2.80$ and $\eta_2 = 0.85$ for the first and second resonator mode, respectively. Calculated polariton frequencies (solid & dotted curves) with distinct resonances labelled. **b** Calculated transmission and identical polariton frequencies, for comparison. **c** Transmission of the 48-QW structure. Coupling strengths: $\eta_1 = 2.83$, $\eta_2 = 0.88$. **d** Calculated transmission and identical polariton frequencies.

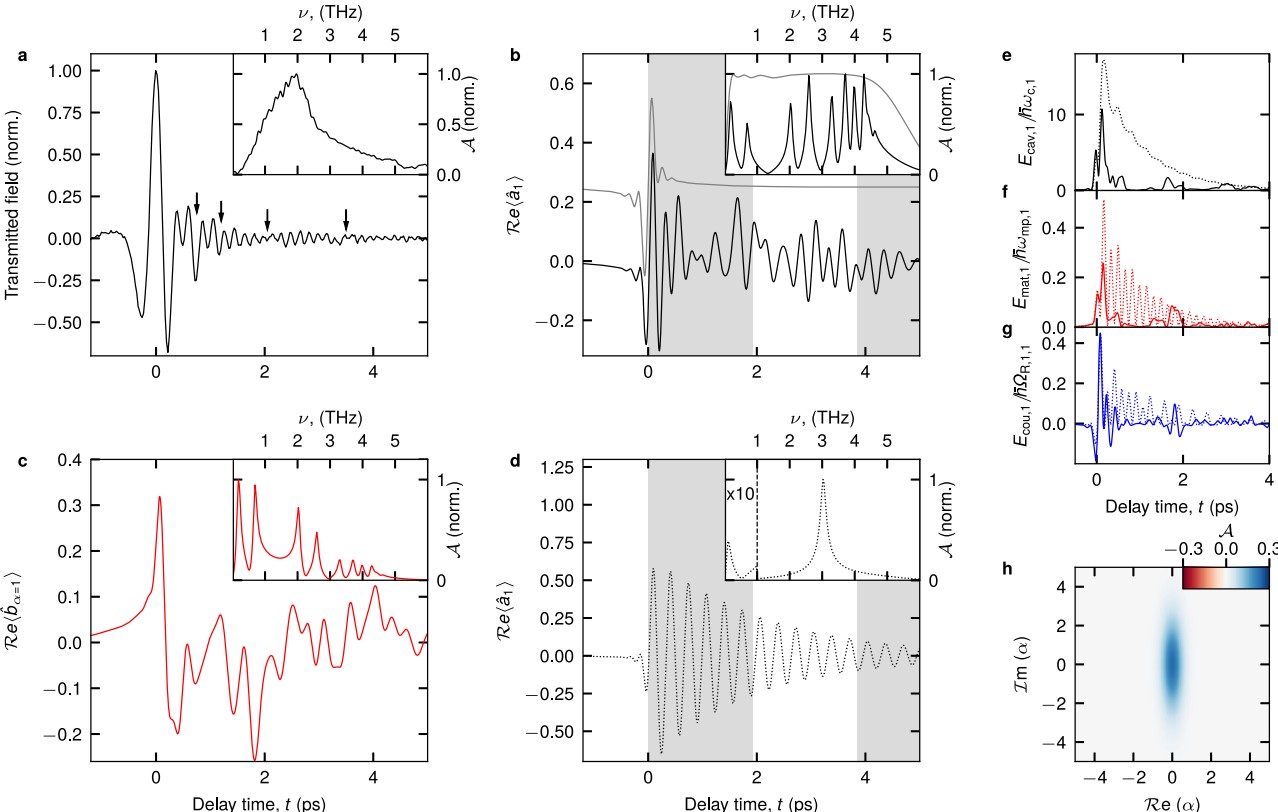

**Fig. 4 | Dynamics and squeezing of extremely strong light-matter coupling.**
**a** Transmitted THz field of the 48-QW sample ($\eta_1 = 2.83$) at $\nu_c = 0.52$ THz (black curve). Inset: Spectral amplitude $\mathcal{A}$ of the THz field. **b** Calculated expectation value for the population of the first cavity mode $Re\langle\hat{a}_{j=1}\rangle$ after excitation (black curve) by a broadband pulse (grey curve). The shading marks one oscillation period of the bare cavity mode. Inset: Corresponding spectra. **c** Calculated expectation value of the polarisation of the first MP mode, $Re\langle\hat{b}_{\alpha=1}\rangle$, and spectrum (inset). **d** $Re\langle\hat{a}_{j=1}\rangle$ for the same coupling strength as in **b**, yet only for a single pair of light and matter modes. Inset: Corresponding spectrum. **e** Energy of the first cavity mode for the full calculation (solid curve) and the single-mode reference (dotted curve) as in panel **d**. **f** Energies of the first MP mode (solid curve) and CR (dotted curve) for the two cases. **g** Corresponding coupling energies between cavity mode and MP mode (solid curve) or CR (dotted curve). **h** Wigner-function representation for the photonic state of a deep-strongly coupled system with $\Omega_R/\omega_c = 3.0$ showing the quasi-probability $\mathcal{A}$ as a function of the field quadratures $Re(\alpha)$ and $Im(\alpha)$.

strongly influence each other, the dominant frequencies in their polarisation reflect the spectral signatures of all MPPs simultaneously – a unique hallmark of multi-mode non-resonant DSC and the strong mixing of all MPs at the same time (see Supplementary Information). In comparison, the dynamics of the cavity field of a single pair of light-matter coupled modes with $\Omega_R^s/\omega_s = 3.19$ is much less structured (Fig. 4d), and its spectrum contains only the lower and upper polariton resonances (Fig. 4d, inset).

An even stronger contrast between the settings of single and multi-mode coupling is evident from the dynamics of the energy redistribution: Whereas energy exchange of a single pair of modes progresses periodically at $\Omega_R^s$ (Fig. 4e–g, dotted curves), here, the large number of participating modes leads to irregularly structured dynamics for the cavity energy, each of the MP modes, and the energy stored in the coupling[28] (Fig. 4e–g, solid curves). Moreover, the extreme nature of the coupling manifests in its vacuum ground state, where strong squeezing of the photonic mode is observed (Fig. 4h). Further characteristics range from a highly non-classical Fock-state probability distribution (see Supplementary Information) to the emission of correlated photon pairs, expected for non-adiabatic modulation of this exotic vacuum ground state[17].

In conclusion, our work represents a leap forward in c-QED by overcoming the limitations of resonant light-matter coupling with our innovative concept of cooperative, multi-mode hybridisation, allowing for a significant boost of light-matter coupling strengths. To this end, we designed a maximally compact resonator metasurface that custom-

tailors multiple plasmon resonances as well as optical modes to form an ultrabroadband spectrum of Landau cavity polaritons covering 6 optical octaves. This extremely strong coupling results in a highly subcycle vacuum energy exchange and a vacuum ground state hosting a record population of 1.17 virtual photons and 1.06 virtual magneto-plasmon excitations. These figures correspond to a coupling strength of $\Omega_R^s/\omega_s = 3.19$ which exceeds previous records of THz c-QED more than twofold. As our structures permit femtosecond optical switching[28], our tenfold increase of the areal density of virtual excitations will highly benefit the detection of vacuum radiation[30,44]. The extremely strong coupling of multiple electronic modes to the common cavity mode even facilitates hybridisation of otherwise orthogonal matter states and can be applied to interactions between a variety of systems such as magnons, phonons or Dirac electrons, including the mixing of entirely different excitations. Combined with the resulting sizeable virtual population of matter and light modes, our concept offers high flexibility and an extensive level of control which can be leveraged, for example, in electronic transport[13,15], light-induced phase transitions[33] or chemical reactions[20–22], by the interaction with vacuum fields.

## Methods

### Semiconductor heterostructure growth and electron-beam lithography

Our semiconductor heterostructures were grown by molecular-beam epitaxy on undoped (100)-oriented GaAs substrates which were

prepared by growing an epitaxial GaAs layer of a thickness of 50 nm followed by an $Al_{0.3}Ga_{0.7}As$/GaAs superlattice to obtain a defect-free, atomically flat surface. The GaAs quantum well (QW) stacks were embedded in $Al_{0.3}Ga_{0.7}As$ barriers, creating rectangular potential wells. Si δ-doping layers were placed symmetrically around the individual QWs, enabling control of the carrier density $\rho_{QW}$ of the two-dimensional electron gases (2DEGs) formed in each QW. The spacing of the QWs and the doping density have been optimised following the strategy discussed in ref. 7. In particular, we took care to exclude coupling between adjacent QWs. Throughout the manuscript, $\rho_{QW}$ is the density per QW, while $\rho_{2D}$ denotes the combined carrier density of the QW stack, integrated over all QWs along the $z$-direction. The QW stacks were capped by a GaAs layer of a thickness of 30 nm (20 nm for the 12-QW sample) for protection against oxidation. Table 1 contains information on the QW and barrier thicknesses.

The THz resonators were fabricated on the surface of the semiconductor structures by electron-beam lithography and wet-chemical processing of polymethylmethacrylat (PMMA) resist followed by thermal vapour-phase deposition of 5 nm of Ti, improving adhesion of the subsequently deposited Au layer of a thickness of 100 nm.

### Wave vector decomposition of resonator near field, and magnetoplasmon modes

The periodicity of our metasurfaces leads to a discretization of the plasmon wave vectors, $|\mathbf{q}_x(\alpha)| = \frac{2\pi}{L_x}\alpha, \alpha \in \mathbb{Z}$, where $L_x$ is the unit cell size of the structure in $x$-direction, and $\alpha$ is the plasmon mode index. For each $\mathbf{q}_x(\alpha)$, the relevant electric field amplitude of the cavity mode is given by the Fourier component $[\mathcal{F}(\mathcal{E}_x)](\mathbf{q}_x, \mathbf{q}_y = 0)$, with $\mathcal{F}(\mathcal{E}_x)$ denoting the 2D Fourier transform of the electric near-field, $\mathcal{E}_x$, in $x$-direction. $\mathcal{E}_x$ is calculated by the finite-element method for the bare resonator on an undoped substrate. The amplitudes are determined separately for each QW. Fig. S3 shows these data for the first two resonator modes, as a function of the wave vector and the depth below the metasurface. For large wave vectors, single-particle excitations become energetically possible and the field amplitude moreover drops sharply, limiting the relevant wave vector spectrum up to a maximum index $|\alpha_c| = 10$.

### Cryogenic THz magneto-spectroscopy

Femtosecond near-infrared pulses (centre wavelength, 807 nm; pulse energy, 5.5 mJ, pulse duration, 33 fs) from a titanium-sapphire amplifier laser (repetition rate, 3 kHz) were used to generate single-cycle THz pulses by optical rectification and to detect the transmitted waveforms by electro-optic sampling. Depending on the required bandwidth, we employed ⟨110⟩-cut ZnTe crystals of a thickness of 1 mm for our structures with up to 12 QWs, and GaP crystals of a thickness of 200 μm for the 24 and 48-QW structures. A mechanical chopper modulated the

THz pulses, allowing for differential detection of the transmitted THz electric field, $E(t)$. The sample was kept at a temperature of 10 K in a magneto-cryostat with a large numerical aperture, enabling magnetic fields up to 5.5 T applied perpendicularly to the sample surface. The recorded electric field was Fourier transformed and referenced to a measurement without a sample in the cryostat to obtain the transmission spectra.

### Sample structure parameters

For each sample, the doping densities, coupling strengths $\eta_j$ for each cavity mode, and the corresponding vacuum photon numbers, $N_j$, are listed in Table 1.

## Data availability

The processed data that support the findings of this study are available at https://epub.uni-regensburg.de/55349/.

## Code availability

The code that support the findings of this study is available from the corresponding author upon request.

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

**Table 1 | Structural and coupling parameters of the 6 QW structures**

| $n_{QW}$ | Doping density $\rho_{QW}$ (cm$^{-2}$) | $t_{QW}$ (nm) | $t_B$ (nm) | $\langle N_1 \rangle$ | $\langle N_2 \rangle$ | $\langle N \rangle$ | $\eta_1$ | $\eta_2$ |
|---|---|---|---|---|---|---|---|---|
| 1 | $1.28 \times 10^{12}$ | 10 | 30 | 0.07 | 0.004 | 0.07 | 0.55 | 0.13 |
| 3 | $7.9 \times 10^{11}$ | 10 | 30 | 0.13 | 0.01 | 0.14 | 0.76 | 0.19 |
| 6 | $1.37 \times 10^{12}$ | 10 | 30 | 0.34 | 0.03 | 0.37 | 1.34 | 0.36 |
| 12 | $1.90 \times 10^{12}$ | 8 | 14 | 0.76 | 0.08 | 0.85 | 2.32 | 0.60 |
| 24 | $2.30 \times 10^{12}$ | 10 | 30 | 0.99 | 0.16 | 1.14 | 2.80 | 0.85 |
| 48 | $1.80 \times 10^{12}$ | 10 | 30 | 1.00 | 0.17 | 1.17 | 2.83 | 0.88 |

$t_{QW}$, $t_B$: Thickness of the QWs and barrier layers, respectively. $\langle N_1 \rangle$, $\langle N_2 \rangle$, $\langle N \rangle$ : vacuum photon numbers for the first and second resonator modes, and for both combined, respectively. $\eta_1$ and $\eta_2$ denote the hypothetical coupling strengths of, in each case, a single pair of light and matter modes with a vacuum photon population of $\langle N_1 \rangle$ or $\langle N_2 \rangle$, respectively, for comparison.

19. Schlawin, F., Kennes, D. M. & Sentef, M. A. Cavity quantum materials. *Appl. Phys. Rev.* **9**, 11312 (2022).
20. Chikkaraddy, R. et al. Single-molecule strong coupling at room temperature in plasmonic nanocavities. *Nature* **535**, 127–130 (2016).
21. Thomas, A. et al. Tilting a ground-state reactivity landscape by vibrational strong coupling. *Science* **363**, 615–619 (2019).
22. Dunkelberger, A. D., Simpkins, B. S., Vurgaftman, I. & Owrutsky, J. C. Vibration-cavity polariton chemistry and dynamics. *Annu. Rev. Phys. Chem.* **73**, 429–451 (2022).
23. Cortese, E. et al. Excitons bound by photon exchange. *Nat. Phys.* **17**, 31–35 (2021).
24. Mornhinweg, J. et al. Tailored subcycle nonlinearities of ultrastrong light-matter coupling. *Phys. Rev. Lett.* **126**, 177404 (2021).
25. Valmorra, F. et al. Vacuum-field-induced THz transport gap in a carbon nanotube quantum dot. *Nat. Commun.* **12**, 5490 (2021).
26. Knorr, M. et al. Intersubband polariton-polariton scattering in a dispersive microcavity. *Phys. Rev. Lett.* **128**, 247401 (2022).
27. Hagenmüller, D., de Liberato, S. & Ciuti, C. Ultrastrong coupling between a cavity resonator and the cyclotron transition of a two-dimensional electron gas in the case of an integer filling factor. *Phys. Rev. B* https://doi.org/10.1103/PhysRevB.81.235303 (2010).
28. Halbhuber, M. et al. Non-adiabatic stripping of a cavity field from deep-strongly coupled electrons. *Nat. Photon.* **14**, 675–679 (2020).
29. Hawking, S. W. Black hole explosions? *Nature* **248**, 30–31 (1974).
30. de Liberato, S., Ciuti, C. & Carusotto, I. Quantum vacuum radiation spectra from a semiconductor microcavity with a time-modulated vacuum Rabi frequency. *Phys. Rev. Lett.* **98**, 103602 (2007).
31. de Liberato, S. Light-matter decoupling in the deep strong coupling regime: the breakdown of the Purcell effect. *Phys. Rev. Lett.* **112**, 16401 (2014).
32. Rajabali, S. et al. Polaritonic nonlocality in light–matter interaction. *Nat. Photon.* **15**, 690–695 (2021).
33. Schlawin, F., Cavalleri, A. & Jaksch, D. Cavity-mediated electron-photon superconductivity. *Phys. Rev. Lett.* **122**, 133602 (2019).
34. Yu, N. et al. Flat optics: controlling wavefronts with optical antenna metasurfaces. *IEEE J. Sel. Top. Quant. Electron.* **19**, 4700423 (2013).
35. Zhou, J., Koschny, T. & Soukoulis, C. M. Magnetic and electric excitations in split ring resonators. *Opt. Express* **15**, 17881–17890 (2007).
36. Chen, H.-T., Taylor, A. J. & Yu, N. A review of metasurfaces: physics and applications. *Rep. Prog. Phys. Phys. Soc.* **79**, 76401 (2016).
37. Chen, H.-T. et al. Complementary planar terahertz metamaterials. *Opt. Express* **15**, 1084–1095 (2007).
38. Olmon, R. L. et al. Optical dielectric function of gold. *Phys. Rev. B* **86**, 235147 (2012).
39. Stern, F. Polarizability of a two-dimensional electron gas. *Phys. Rev. Lett.* **18**, 546–548 (1967).
40. Popov, V. V., Polischuk, O. V. & Shur, M. S. Resonant excitation of plasma oscillations in a partially gated two-dimensional electron layer. *J. Appl. Phys.* **98**, 33510 (2005).
41. Kushwaha, M. S. Plasmons and magnetoplasmons in semiconductor heterostructures. *Surf. Sci. Rep.* **41**, 1–416 (2001).
42. Hopfield, J. J. Theory of the contribution of excitons to the complex dielectric constant of crystals. *Phys. Rev.* **112**, 1555–1567 (1958).
43. Zhang, Q. et al. Superradiant decay of cyclotron resonance of two-dimensional electron gases. *Phys. Rev. Lett.* **113**, 47601 (2014).
44. Garziano, L., Ridolfo, A., Stassi, R., Di Stefano, O. & Savasta, S. Switching on and off of ultrastrong light-matter interaction: photon statistics of quantum vacuum radiation. *Phys. Rev. A* **88**, 63829 (2013).

## Acknowledgements
The authors thank Dieter Schuh and Imke Gronwald for valuable discussions and technical support. We gratefully acknowledge support by the Deutsche Forschungsgemeinschaft through Project IDs 231111959 and 422 31469 5032-SFB1277 (Subproject A01), grants no. LA 3307/1-2, and HU 1598/8.

## Author contributions
J.M., M.H, D.B., R.H. and C.L. conceived the study. M.P., M.H. and D.B. designed, realised and characterised the semiconductor heterostructures. J.M., M.H. and L.K.D. modelled the metasurfaces and fabricated the samples with support from T.I., J.R., D.B. and C.L.; J.M., M.H., L.K.D. and J.R. carried out the experiments with support from R.H. and C.L. The theoretical modelling was carried out by J.M., R.H. and C.L.; D.B., R.H. and C.L. supervised the study. All authors analysed the data and discussed the results. J.M., R.H. and C.L. wrote the manuscript with contributions from all authors.

## Funding

## Competing interests
The authors declare no competing interests.
