## [Peer Review File · Nature Communications]

REVIEWER COMMENTS

Reviewer #1 (Remarks to the Author):

This work by Mornhinweg et al. describes the design, fabrication and characterization of a metasurface working in the THz spectral range and exhibiting an extremely strong degree of coupling between photonic and electronic states. The authors achieved this remarkable result by coupling an especially designed plasmonic resonator with very low mode volume to the Landau resonances of an underlying quantum well. The outstanding feature of these systems is that strong, multimode off-resonant coupling was predicted and observed, leading to a control of the Rabi frequency that can be basically achieved just by controlling the dipole resonator strength.

The work is carried out with great rigor and completeness, both from the experimental and theoretical points of view: a very elegant metasurface design was proposed and implemented, and a full, step-by-step coupling to an increasingly deep quantum well stack was obtained and characterized, generating solid data to support the authors' claims. The results are very interesting and bear a strong potential impact in the field of low energy excitations. I recommend the manuscript for publications after very minor revisions.

Here are a few minor queries/curiosities:

- Page 6, lines 143-145 and SI page 11, eq 13: can the authors attempt to give a physical meaning to the common scaling factor applied to all Rabi frequencies?

- Figure 3c: a spectral feature between LP2 and UP1,0 is visible as a bifurcation of the transmittance for high CR frequencies. This is not accounted for in the calculation reported in Figure 3d. Can the authors comment on this?

- The authors model the effect of an applied magnetic field with the formation of Landau-quantized cyclotron resonances in the quantum well structure. However, similar effects (albeit unquantized in most regimes) are also induced in the metal supporting plasmon resonances (10.1103/PhysRevLett.104.147401, doi.org/10.1063/5.0050034, 10.1021/acs.jpcc.1c09900). As far as I understand, the authors have not considered this point. In fact, the effect will be quantitatively very small or negligible since the cyclotron resonance scales with the inverse of the effective mass of the charge carrier and GaAs has carriers with a much smaller effective mass (~ 0.01) than those of gold. Nevertheless, for completeness this point should probably be addressed.

Reviewer #2 (Remarks to the Author):

The manuscript introduces an original approach to strong light-matter coupling, stepping away from the traditional resonant interactions to achieve remarkably high coupling strengths. Key findings include:

- Multi-Mode Strong Coupling with magnetoplasmons: The research explores a new realm of strong light-matter interaction by coupling multiple highly non-resonant magnetoplasmon modes to various optical modes. This creates an ultrabroadband spectrum of over 20 polaritons spanning several optical octaves.
- Record-Breaking Coupling Strength: The proposed method achieves a record coupling strength of 3.19, surpassing previous values almost twofold. This high coupling strength allows for strong subcycle exchange of vacuum energy between multiple modes.
- Non-Resonant Multi-Mode Coupling: Unlike previous methods that focused on resonant coupling between a single photonic and a single electronic mode, this approach involves the coupling of multiple non-resonant plasmon modes simultaneously and cooperatively to several optical modes of a metallic metasurface.
- Metasurface Design: The authors custom-tailor the plasmon quantum states using a compact resonator metasurface, which significantly reduces the cavity size as compared to previous approaches while providing a large near-field enhancement.
- Subcycle Vacuum Energy Exchange: The research demonstrates non-trivial subcycle exchange of vacuum energy between optical and electronic modes, which is a novel phenomenon with applications in various fields, potentially strong-field physics.

The experimental data are of high quality and the manuscript is generally well written. The results open up new possibilities for non-adiabatic cavity quantum electrodynamics phenomena, such as vacuum-field modified transport, control of cavity chemistry, and the observation of phenomena like Unruh-Hawking radiation.

I would recommend publication in Nature Communications.

However, I would have some remarks that the authors should address:

- The vacuum population is calculated and not measured. The authors should reduce a bit the emphasis on that
- The contour plots of the spectra are nice, but some cuts of spectra with multiple peaks and their lineshapes should be displayed in the manuscript
- Concerning the platform in Ref. 13, the more recent paper Appugliese et al., Science 375, 1030 (2022) is probably more relevant.

Reviewer #3 (Remarks to the Author):

This is review report of “Mode-multiplexing deep strong light matter interactions” authored by Mornhinweg et al. This work deals with strong light matter interactions mediated by complimentary metastructures. Authors have demonstrated experimental verifications supported by theoretical backgrounds of the tunable magnetoplasmons. Work is fundamental, timely and also interesting to the Terahertz, metamaterial, THz spintronic and broadly applied physics communities. However, I have few concerns regarding this work, authors can answer them.

1) My first concern is regarding the metasurface designs. Why they have chosen a complimentary structure instead of typical designs. In fact, typical designs can lead to much stronger field confinements at the gaps. Did they try for typical split gap type of designs? Curious to know this.

2) My second concern is, how the 2 DEG is created?? What is the QW design? How did they optimize them. Its know that 2DEG can be created in multiple QW or even in single QW, for example, in RTD like structures. However, such details are missing. I feel those are very important.

3) Can authors provide effective bandstructures of the QW? How such bands are changing with increasing number of layers.

4) Electric field distribution in the -Z direction is important in this work. I am little doubtful about the saturation in magnetoplasmon behavior. Is it because of field distributions or multiple QWs. This point probably needs to discuss.

5) I don't agree that resonance at 1.95 THz is dipolar mode or 2nd order mode as claimed by the authors. It's actually much higher order mode. In order to excite 2nd order mode one needs to change the polarization. Please rectify that. Please check Soukoulis's paper or publications originated from LANL in last decade.

Minor comments,

Please improve figure qualities. It will help to reach broader audience.

Overall, work is good and can be considered further.

Nature Communications Manuscript NCOMMS-23-08801-T
Response to Reviewer comments

Reviewer #1 (Remarks to the Author):

Referee: This work by Mornhinweg et al. describes the design, fabrication and characterization of a metasurface working in the THz spectral range and exhibiting an extremely strong degree of coupling between photonic and electronic states. The authors achieved this remarkable result by coupling an especially designed plasmonic resonator with very low mode volume to the Landau resonances of an underlying quantum well. The outstanding feature of these systems is that strong, multimode off-resonant coupling was predicted and observed, leading to a control of the Rabi frequency that can be basically achieved just by controlling the dipole resonator strength.

The work is carried out with great rigor and completeness, both from the experimental and theoretical points of view: a very elegant metasurface design was proposed and implemented, and a full, step-by-step coupling to an increasingly deep quantum well stack was obtained and characterized, generating solid data to support the authors' claims. The results are very interesting and bear a strong potential impact in the field of low energy excitations. I recommend the manuscript for publications after very minor revisions.

Response 1: We thank the Referee very much for their time, the extremely positive review and the very kind words.

Referee:

Here are a few minor queries/curiosities:

- Page 6, lines 143-145 and SI page 11, eq 13: can the authors attempt to give a physical meaning to the common scaling factor applied to all Rabi frequencies?

Response 2:

The common scaling factor can be cast as $E_j/\sqrt{V_j}$, whereby E_j is the electric field amplitude of the j^{th} resonator mode and $1/\sqrt{V_j}$ is the inverse root of the mode volume, since the coupling strength scales with both of these factors [Maissen et al., Phys. Rev. B **90**, 205309 (2014)]. Our classical electro-dynamical simulations reproduce the coupling strengths and thus determine $E_j/\sqrt{V_j}$ indirectly, without free fit parameters. We agree with the Referee that it would be beneficial to attribute the scaling factor with E/\sqrt{V} , and we have modified the manuscript accordingly (**change #1**).

Referee:

- Figure 3c: a spectral feature between LP2 and UP1,0 is visible as a bifurcation of the transmittance for high CR frequencies. This is not accounted for in the calculation reported in Figure 3d. Can the authors comment on this?

Response 3:

The additional mode pointed out by the Referee results from coupling of x and y-polarized resonator modes through the cyclotron resonance and its gyrotropic polarization response,

which becomes more and more dominant as the static magnetic bias field and hence the cyclotron resonance frequency increase. This coupling is fully accounted for in the electro-dynamical simulations based on solving Maxwell's equations, previously not shown in the Supplemental Material document for the 24 and 48-QW structures. Since the coupled mode in question i) couples to the x polarization probed in our experiment only indirectly, i.e., through the cyclotron resonance, and ii) involves a cavity mode with a rather delocalized field distribution, its overall effect is weak. Moreover, the modes of our resonator with the strongest field enhancement couple to the x component of the far field, by design. Any light-matter coupling from y-polarized modes are thus much smaller, such that they are visible only in our most strongly coupled structures.

Thus, in order to keep the theory straightforward, we decided to not treat the coupling of y-polarized modes in our quantum model. We have added a corresponding remark in the manuscript as well as graphs to the Supplementary Material document (**change #2**).

Figure S13 a, THz magneto-transmission of the 24-QW sample as a function of ν_c (see Fig. 3). Calculated polariton frequencies (solid & dashed curves) with distinct resonances marked. **b**, FEFD simulated transmission and polariton frequencies. **c**, Transmission of the 48-QW structure. **d**, FEFD simulated transmission and identical polariton frequencies.

Referee:

The authors model the effect of an applied magnetic field with the formation of Landau-quantized cyclotron resonances in the quantum well structure. However, similar effects (albeit unquantized in most regimes) are also induced in the metal supporting plasmon resonances (10.1103/PhysRevLett.104.147401, doi.org/10.1063/5.0050034, 10.1021/acs.jpcc.1c09900). As far as I understand, the authors have not considered this point. In fact, the effect will be quantitatively very small or negligible since the cyclotron resonance scales with the inverse of the effective mass of the charge carrier and GaAs has carriers with a much smaller effective mass (~ 0.01) than those of gold. Nevertheless, for completeness this point should probably be addressed.

Response 4:

The Referee raises an interesting point regarding the cyclotron resonance of the resonator material, usually not considered in the THz c-QED community. As he/she points out, the large value of the effective electron mass m^* in gold near the Fermi level (approximately $m^*/m_0 = 1$ [<https://journals.aps.org/pr/abstract/10.1103/PhysRev.136.A1383>], where m_0 is the free electron mass) leads to a much lower cyclotron resonance frequency as for our GaAs QWs when exposed to the same magnetic field. More precisely, at $\nu_c = 1$ THz for our QWs ($B = 2.5$ T), the CR of gold is $\nu_{c,Au} \sim 70$ GHz. At these frequencies, which even lie quite significantly below our measurement bandwidth with a lower limit of ~ 200 GHz, no significant coupling to the relevant modes of our structures has to be expected. We have added a brief corresponding statement (**change #3**).

Reviewer #2 (Remarks to the Author):

The manuscript introduces an original approach to strong light-matter coupling, stepping away from the traditional resonant interactions to achieve remarkably high coupling strengths. Key findings include:

- Multi-Mode Strong Coupling with magnetoplasmons: The research explores a new realm of strong light-matter interaction by coupling multiple highly non-resonant magnetoplasmon modes to various optical modes. This creates an ultrabroadband spectrum of over 20 polaritons spanning several optical octaves.
- Record-Breaking Coupling Strength: The proposed method achieves a record coupling strength of 3.19, surpassing previous values almost twofold. This high coupling strength allows for strong subcycle exchange of vacuum energy between multiple modes.
- Non-Resonant Multi-Mode Coupling: Unlike previous methods that focused on resonant coupling between a single photonic and a single electronic mode, this approach involves the coupling of multiple non-resonant plasmon modes simultaneously and cooperatively to several optical modes of a metallic metasurface.
- Metasurface Design: The authors custom-tailor the plasmon quantum states using a compact resonator metasurface, which significantly reduces the cavity size as compared to previous approaches while providing a large near-field enhancement.
- Subcycle Vacuum Energy Exchange: The research demonstrates non-trivial subcycle exchange of vacuum energy between optical and electronic modes, which is a novel phenomenon with applications in various fields, potentially strong-field physics.

The experimental data are of high quality and the manuscript is generally well written. The results open up new possibilities for non-adiabatic cavity quantum electrodynamics phenomena, such as vacuum-field modified transport, control of cavity chemistry, and the observation of phenomena like Unruh-Hawking radiation.

I would recommend publication in Nature Communications.

Response:

We thank the Referee very much for the time they invested in examining our work and for the very positive review and recommendation for publication. Below, we address the Referee's remarks.

Referee:

However, I would have some remarks that the authors should address:

- The vacuum population is calculated and not measured. The authors should reduce a bit the emphasis on that

Response 1:

Measuring the vacuum population represents a significant and ongoing challenge, such that the c-QED community tends to state the vacuum population without further mentioning that it is, thus far, a calculated quantity. We agree though with the Referee and we have made it clear that the stated vacuum population is calculated, wherever this was not already clear from the text (**change #4**).

Referee:

- The contour plots of the spectra are nice, but some cuts of spectra with multiple peaks and their lineshapes should be displayed in the manuscript

Response 2:

We agree that some aspects of the data such as line shapes are better visible in one-dimensional plots. We have added waterfall plots for a selection of the measured data in the Supplementary Materials document (**change #5**).

Figure S15 | Waterfall plots of Figure 2. a, THz magneto-transmission as a function of ν_c of the single-QW structure. **d**, Spectrum obtained from time-domain quantum model **b**, Transmission of the 3-QW structure and **e**, simulation. **c** Transmission of the 6-QW structure, and **f**, simulation. **d**, Transmission of the 12-QW structure, and **g**, simulation.

Figure S16 | Waterfall plots of the data of Fig. 3. a, THz magneto-transmission of the 24-QW sample as a function of ν_c . **b,** Corresponding calculated transmission. **c,** Transmission of the 48-QW structure. **d,** Corresponding calculated transmission.

Referee:

- Concerning the platform in Ref. 13, the more recent paper Appugliese et al., Science 375, 1030 (2022) is probably more relevant.

Response 3:

We thank the Referee for pointing out this relevant reference, which we have added to the manuscript (**change #6**).

Reviewer #3 (Remarks to the Author):

This is review report of “Mode-multiplexing deep strong light matter interactions” authored by Mornhinweg et al. This work deals with strong light matter interactions mediated by complimentary metastructures. Authors have demonstrated experimental verifications supported by theoretical backgrounds of the tunable magnetoplasmons. Work is fundamental, timely and also interesting to the Terahertz, metamaterial, THz spintronic and broadly applied physics communities. However, I have few concerns regarding this work, authors can answer them.

Response 1:

We thank the Referee for their time spent on assessing our manuscript and their very positive review. Below, we address their questions.

Reviewer:

1) My first concern is regarding the metasurface designs. Why they have chosen a complimentary structure instead of typical designs. In fact, typical designs can lead to much stronger field confinements at the gaps. Did they try for typical split gap type of designs? Curious to know this.

Response 2:

Our resonator is the result of several steps of optimization, simultaneously targeting i) strong electric field enhancement, ii) plasmon confinement functionality, and iii) a maximally compact footprint. While the starting point of our design is a structure that would certainly be classified as inverted, this attribution is much less clear for the compacted version, where practically all of the unstructured metallization was removed. Moreover, we point out that the maximum field enhancement by itself does not fully determine the coupling strength of planar c-QED resonator structures. One further important factor is the area of field enhancement, with which the number of electrons collectively coupling to the resonator mode scales. We do not state that our resonators lead to the highest possible coupling strengths achievable in principle, but from all the resonator design variations tested over the course of our study, the presented one is the most advanced. We have added a short remark in the manuscript (**change #7**).

Reviewer:

2) My second concern is, how the 2 DEG is created?? What is the QW design? How did they optimize them. Its know that 2DEG can be created in multiple QW or even in single QW, for example, in RTD like structures. However, such details are missing. I feel those are very important.

3) Can authors provide effective bandstructures of the QW? How such bands are changing with increasing number of layers.

Response 3:

We thank the referee for highlighting the importance of the 2DEG design. Our quantum wells (QWs) were grown by molecular beam epitaxy on undoped (100)-oriented GaAs substrates. The wells were embedded in $\text{Al}_{0.3}\text{Ga}_{0.7}\text{As}$ barrier layers for electronic confinement, forming rectangular potential wells. Silicon δ -doping layers were placed symmetrically around each QW to control the electron doping concentration. The spacing of the QWs was optimized following the strategy of ref. 7 [Bayer, A. et al., Nano Lett. **17**, 6340 (2017)]. In particular, it was (i) minimized to achieve the highest possible cumulated charge carrier density within the near-field volume of our resonator structures while (ii) preventing coupling of adjacent wells. The wells are thus independent of each other as far as the electronic band structure is concerned,

which is thus represented by the quadratic dispersion of the GaAs conduction band in the QW plane, and a discretized energy spectrum for the quantized out-of-plane component of the k vector.

We have expanded our description of the semiconductor heterostructures in the revised version of the manuscript and stated that we exclude mutual electronic coupling between individual QWs. Furthermore, we have added structural QW parameters to table 1 (**change #8**).

Reviewer:

4) Electric field distribution in the $-Z$ direction is important in this work. I am little doubtful about the saturation in magnetoplasmon behavior. Is it because of field distributions or multiple QWs. This point probably needs to discuss.

Response 4:

The Referee is right that the z -distribution of the near-field is central to optimizing the coupling strength. A straightforward explanation of the saturation behavior considers the asymptotically exponential decay, in z direction, of the near field of the empty cavity, which limits the total number of QWs that can contribute to light-matter coupling for reasons of geometrical overlap. A more sophisticated analysis accounts for (i) superradiant coupling of the polarization of the cyclotron resonance [Q. Zhang et al, Phys. Rev. Lett. **113**, 047601 (2014)] in our highly doped, magnetically biased QWs and the related (ii) shaping of the vacuum mode by the matter component [A. Bayer et al., Nano Lett. **17**, 6340 (2017)]. While these effects lead to an effective radiative coupling of QWs which would otherwise be out of reach of the near-field of the empty cavity, it also increases the effective mode volume, which then leads to a decrease of the coupling strength. Combining both effects, however, the coupling strength generally increases for an increasing number of QWs, but with diminishing returns.

We point out that this mechanism is fully accounted for in our finite-element simulation which implements the cyclotron resonance by a dielectric tensor. Moreover, it does not influence our Hamiltonian quantum model. We have added a corresponding statement in the revised version of the manuscript (**change #9**).

Reviewer:

5) I don't agree that resonance at 1.95 THz is dipolar mode or 2nd order mode as claimed by the authors. It's actually much higher order mode. In order to excite 2nd order mode one needs to change the polarization. Please rectify that. Please check Soukoulis's paper or publications originated from LANL in last decade.

Response 5:

The Referee argues that the energy order of all modes of our resonator structure starts with a x -polarized mode and continues with a y -polarized second-order mode, whereby the mode we named "dipolar mode" is of higher order. We agree that in principle, the mode second-lowest in energy may have a different polarization than the fundamental one.

For the terminology, we adhered to the community's standard [4,5,7,8,16,24,28,32] of designating the mode involving capacitive and inductive elements as "LC". Moreover, we termed the second mode where the field distribution is not tied to these structures "dipolar". Furthermore, we enumerated only the modes which couple to the x -polarized far-field components. While this enumeration may not sort all resonator modes by energy in ascending order, we confirmed that this is a common strategy (e.g., J. Zhou, Opt. Express **15**, 17881 (2007), Soukoulis group).

However, we see the Reviewer's point that the term "dipolar mode" is not an optimum choice here. We have thus replaced it by "higher-order mode" throughout all documents, clarified that our enumeration concerns x -polarized modes, and added the above reference (**change #10**).

Reviewer:

Minor comments, Please improve figure qualities. It will help to reach broader audience. Overall, work is good and can be considered further.

Response:

We realize that the figures in the document provided to the Reviewers were not of optimum quality, which we believe to be a purely technical issue arising in the conversion process. We are ready to provide high-quality versions as necessary.

REVIEWERS' COMMENTS

Reviewer #1 (Remarks to the Author):

From my point of view the revised version of this manuscript is suitable for publication in Nature Communications.

Reviewer #2 (Remarks to the Author):

I am very satisfied by the responses provided by the authors and by their revisions. Hence, I can fully recommend the publication of this manuscript in Nature Communications.

Reviewer #3 (Remarks to the Author):

Authors have commented my queries satisfactorily. This is indeed a nice piece of work. I recommend acceptance.